# Country Parks as Sites of Emergency Medical Facility: A Case of COVID-19 in Hangzhou, China

**DOI:** 10.3390/ijerph192315876

**Published:** 2022-11-29

**Authors:** Shuai Li, Zheng Wu, Jiefang Tang, Shuo Wang, Pengfei Wang

**Affiliations:** 1College of Landscape Architecture and Art, Henan Agricultural University, Zhengzhou 450002, China; 2School of Public Policy & Management, China University of Mining and Technology, Xuzhou 221000, China

**Keywords:** COVID-19, public health emergency, emergency medical facility, country park, site selection

## Abstract

City parks are suitable sites for the construction of emergency medical facilities. A comparison of various types of city parks revealed that country parks fit closely with site selection conditions for emergency medical facilities. Based on the latter site selection requirements, eight impact factors such as park type, effective avoidance area, spatial fragmentation degree, water source protection area, wind direction, distance from city center, impermeability, and transport duration were quantified, and then 29 country parks in the Hangzhou Urban Area were compared using Principal Component Analysis (PCA). The calculation results showed that Linglong Country Park has the highest score, taking into account the characteristics of safety, scalability, rehabilitation, convenience, pollution prevention, and isolation. Linglong can be given priority selection as a target location for emergency medical facilities. In addition, Silver Lake Country Park, Dongqiao Country Park, Taihuyuan Country Park, and Tuankou Country Park have higher scores and can be used as alternative targets for emergency plans. The scoring results prove that the evaluation method has a high degree of rigor, a significant degree of discrimination, and a high degree of consistency between the validity and weight assignment of each impact factor. In view of the different geographical conditions in each region, the weight assignment of each impact factor can be adjusted according to local conditions and can help make effective use of existing conditions and avoid disadvantages.

## 1. Research Background

### 1.1. Research Status

In order to fight against SARS, the Chinese government established the Beijing Xiaotangshan hospital in 2003 and created a new model of emergency medical facilities [1]. By 2020, when the new crown pneumonia (COVID-19) epidemic broke out, Wuhan followed the “Beijing Xiaotangshan Model” to build Huoshenshan, Leishenshan hospital as an emergency medical facility [2]. In addition, due to the impact of COVID-19, a total of 39 emergency medical facilities under the “Beijing Xiaotangshan Model” had been built across China.

As the epidemic spreads around the world, some countries had also begun to attempt the rapid construction of emergency medical facilities through standard module series prefabricated structures, such as Russia [3], Italy [4], South Korea [5], Iran [6], etc. This fully reflects the practicality, timeliness, and effectiveness of the “Beijing Xiaotangshan Model”. With the advantages of mobility and other advantages, the rapid construction of emergency medical facilities has become an important means of epidemic treatment and prevention and control. The safety of site selection and the high efficiency of construction are the prerequisites and guarantees for emergency medical facilities to play the role of rescue [7].

In the 2003 SARS and 2020 COVID-19 pandemics, Beijing twice started the construction of Xiaotangshan hospital for anti-epidemic measures [8]. During the first preparation, the scientificity of site selection was considered. Xiaotangshan Hospital is located in Xiaotangshan, east of Changping District, Beijing and close to Xiaotangshan Sanatorium, with good infrastructure conditions; the distance between the hospital and the nearest settlement in Xiaotangshan is more than 500 m, and between the nearest waters and Jing-Mi diversion canal is more than 4000 m (Figure 1). Based on the above conditions, the infectious disease hospital can not only be spatially isolated from the surrounding environment, waters, and population, but must also provide space for the expansion of facilities in the short term. However, the site selection is also questionable. First of all, from May to August 2003, the dominant wind direction in Beijing was the east wind. Xiaotangshan is located in the east of Changping District and belongs to the upwind area, which is not conducive to the diffusion of waste gas. Secondly, Xiaotangshan used to be the palace of the Ming and Qing Dynasties [9]. The south side of the site is the site of the imperial pool of the Qing palace, which is not conducive to the protection of cultural relics and historic sites. By 2020, when the hospital was rebuilt, the hospital environment has changed greatly. Residential areas with high-rise buildings have been built on the east and west sides of the target site, especially the distance between the residential area on the east side and the hospital, which is less than 30 m (Figure 2) and violates the original intention of selecting emergency medical facilities away from densely populated areas.

Based on the China’s Guidelines for Disaster Prevention and Avoidance of Urban Green Space [10], which stipulates that urban disaster prevention green spaces should include long-term (more than 30 days) risk-avoidance green space, and clearly states that “long-term risk-avoidance green space… generally combined with regional green space such as country parks”. Anson et al. focused on the positive role played by country parks during the outbreak [11]. Tang Jiefang et al. [12] proposed that Suburban Forest Park has a high degree of compatibility with the location of emergency medical facilities and demonstrated the former is an ideal type of epidemic avoidance green space; Li Liang et al. [13] proposed the transformation method of infrastructure such as traffic, seepage prevention, drainage, etc., based on the advantages of low construction cost, sufficient space, and easy isolation of parks and green spaces on the edge of the city. Ou Yangdong et al. [14] proposed to incorporate the emergency treatment facilities of public health emergencies into the land space planning system, and country parks can be used as temporary special treatment facilities construction sites. Li Shuai et al. [15] initially established a quantitative evaluation system for the location of emergency medical facilities. While Lu Xun et al. [16] promoted the planning and construction of country park infectious disease hospitals to build urban epidemic prevention barriers. In New York City, the emergency hospital had been constructed inside Central Park [17]. In Madrid City, the Exhibition Center was transformed into a temporary hospital [18]. In Mulhouse City, the hospital parking lots were used as the site for a field hospital [19]. The site of emergency medical facilities immediately became the focus of epidemic prevention and control. These site conditions of emergency medical facilities are uneven, and the scientificity remains to be discussed. Improper site could lead to severe secondary transmission.

### 1.2. Overview of the Research Site

As a central city in the Yangtze River Delta (Figure 3), Hangzhou has the characteristics of large population, high density, and strong mobility. As of the end of 2019, Hangzhou has a permanent population of 10.36 million, of which the floating population is as high as 12.043 million, ranking eighth among Chinese cities. In addition, Hangzhou is also a national historical and cultural city; in 2019, the number of tourists to Hangzhou reached 208.14 million. In addition, Hangzhou is an important transportation hub in Southeast China, with diversified and dense passenger and freight transportation routes. As of the end of 2019, the total passenger traffic of Hangzhou, including railways, civil aviation, highways, and waterways, reached 208.88 million passengers, and the total freight volume reached 363.84 million tons [20]. In response to the global spread of COVID-19, regional gateway cities have become more likely to be the place where the epidemic is imported and outbreaks [21]. At the initial stage of the epidemic outbreak in Hangzhou in 2022, 64 locally confirmed cases were reported, but the social impact of the epidemic was relatively broad, resulting in: the area of the closed control area was 10 square kilometers, and about 10,000 people were in the closed control area; the control area is 45 square kilometers, with about 140,000 people in the control area; the prevention area is 360 square kilometers, with about 278,000 people in it. The huge population size and high-frequency flow of people are hotbeds for the rapid spread of the epidemic, and diverse transportation conditions also provide a convenient way for the rapid spread of the epidemic. Therefore, it is imperative to make corresponding emergency plans.

## 2. Research Objects and Methods

The Hangzhou Master Plan (2001–2020), revised in 2016, stipulates that: “the goal is to create an ecological city, … relying on natural factors such as farmland, rivers and natural mountains, 6 country parks are set up between the urban area and multiple clusters”. In 2019, Hangzhou City’s urban greening work pointed out that: start the “Hundred Park Construction” action, and the city will complete the construction of 100 various parks, including country parks, within five years. The Hangzhou Municipal Bureau of Landscape and Cultural Relics put forward in the “recommendations on further enhancing the planning and construction of urban parks in Hangzhou city”: “Building a balanced urban-rural park system, consisting of country parks, urban parks…”. The development plan of Hangzhou Country Park provides necessary construction sites for the site selection of emergency medical facilities.

### 2.1. Research Objects and Data Sources

This paper takes the administrative boundary of the Hangzhou urban area as the research scope, including a land area of 8289 km^2^ [22]. The list of country parks is based on the 29 country parks defined in the “Research on Hangzhou Country Park Planning System from Protection and Utilization to Mechanism Management” [23] (Table 1), using the registration and digitization functions of ArcGIS 10.5, to correct the location, area and other information of 29 country parks (Figure 4).

### 2.2. Research Methods

Firstly, in accordance with the requirements of “The Design Guidelines for Hospitals for Patients with SARS” [24], “The Design Standard of Infectious Disease Emergency Medical Facilities for Novel Coronavirus Pneumonia” [25], and “Guidelines for site selection, design, construction and operation management of emergency infectious disease hospitals” [26] promulgated by the Chinese government, and combined with the location characteristics of emergency medical facilities and the design principles of infectious disease hospitals, the following eight influencing factors are taken as variables:

Park type (*f*_1_): country parks are divided into different categories according to different landscape characteristics, surface morphology, and resource types;

Effective avoidance area (*f*_2_): according to the definition of effective avoidance area in “The design guideline of urban disaster-prevention green space” [27], the effective avoidance area in country parks refers to the total area of the park minus the water area, transportation, buildings and their falling objects, the area affected by the collapse and other areas, which can be used for the construction of emergency medical facilities;

Spatial fragmentation degree (*f*_3_): spatial patchiness is the most common form of landscape pattern, which reflects the heterogeneity of the landscape; landscape fragmentation represents the fragmentation degree of the park divided by waters, roads, and structures, etc., and reflects the complexity of the park’s spatial structure [28];

Water source protection area (*f*_4_): a certain range designated to prevent water source pollution and ensure water quality, and requiring special protection;

Wind direction (*f*_5_): the direction of stroke in the city. Since the dominant wind direction in the city is not obvious, the impact of wind direction on site selection is expressed by quantifying the wind frequency in all directions;

Distance from the city center (*f*_6_): distance from each country park to the city center;

Impermeability (*f*_7_): the buried depth of groundwater refers to the buried depth of underground diving, that is, the distance from the surface of the groundwater to the surface of the ground; the permeability coefficient is a quantitative index to measure the permeability of the groundwater in the site, And the ratio of groundwater depth to permeability coefficient can be used as an index to measure the degree of water-resisting property in the target area [29];

Transport duration (*f*_8_): Time required to transfer COVID-19 patients from designated hospitals in the urban area to emergency medical facilities.

Secondly, the impact of each influencing factor on the location of emergency medical facilities is different. This study uses PCA to assign the weight of eight influencing factors. Due to the different measurement scales of the original data of each influencing factor, the original data needs to be standardized during the principal component analysis, and then the SPSS 26.0 software is used to calculate the characteristic values of the principal components of each indicator (λ), Initial factor load (*f_i_*), variance percentage (%) and principal component (F), establish the principal component model, obtain the index coefficient in the comprehensive model, and finally conduct normalization to obtain the weight value [30]. 

The principal component analysis of 29 groups of country park data shows that the KMO statistic is 0.649, greater than 0.5, and the p value of Bartlett’s sphericity test is 0, less than 0.05, indicating that there is a certain correlation between variables, which can be used for principal component analysis. In order to retain the information of the original data as much as possible, three principal components with eigenvalues greater than 1 are extracted, and the cumulative contribution rate reaches 77.311%, which can basically reflect most of the information of the original data. The initial factor load, eigenvalue, and contribution rate of each principal component are shown in Table 2 and Table 3.

The principal component weight model is calculated from Table 2 and Table 3, and then the weighted average and normalization are performed according to the variance contribution rate to obtain the weight values of each influencing factor:*F* = 0.0964*f*_1_ + 0.1466*f*_2_ + 0.1813*f*_3_ + 0.1607*f*_4_ + 0.0971*f*_5_ + 0.0852*f*_6_ + 0.0858*f*_7_ + 0.1469*f*_8_

That is, the corresponding weights of park type, effective avoidance area, spatial fragmentation degree, water source protection area, wind direction, distance from the city center, impermeability, and transport duration are *w*_1_ = 0.0964, *w*_2_ = 0.1466, *w*_3_ = 0.1813, *w*_4_ = 0.1607, *w*_5_ = 0.0971, *w*_6_ = 0.0852, *w*_7_ = 0.0858, and *w*_8_ = 0.1469, respectively.

Thirdly, all impact factors were assigned using a five-level system and were divided into five levels from inferior to excellent, and the corresponding scores were 20, 40, 60, 80, and 100 points. According to the actual situation of each country park, the Natural Breaks (Jenks) was used to assign a value to each impact factor, and then the value score (*f_i_*) was multiplied by the corresponding weight (*w_i_*) to obtain the score (*y_i_*) of the impact factor, and then the score of each impact factor (*y_i_*) was obtained, and the impact factor score was weighted and calculated to obtain the comprehensive score (*Y*) of the country park. The formula is as follows:(1)$$\sum_{i=1}^{8}w_{i}=1$$
(2)$$y_{i}=w_{i}f_{i}$$
(3)$$Y=\sum_{i=1}^{8}w_{i}f_{i}$$
(4)$$\begin{array}{l} Y=0.0964f_{1}+0.1466f_{2}+0.1813f_{3}+0.1607f_{4}+0.0971f_{5}\\ +0.0852f_{6}+0.0858f_{7}+0.1469f_{8} \end{array}$$

Fourth, the comprehensive scores of 29 country parks were calculated and ranked in sequence, and the country parks with the high scores are more suitable as the site selection targets for emergency medical facilities.

## 3. Impact Factor Assignment Method

### 3.1. Park Type (f*_1_*)

Considering the landscape characteristics, surface morphology, and resource types of country parks, the 29 country parks could be divided into five types: wetland park, heritage park, mountain park, pastoral park, and forest park.

Grade I: water intake or contact with pathogenic microorganisms is an important vector of diseases, such as cholera, hepatitis A, and typhoid fever, which can be transmitted by water. Among the 37 legal infectious diseases in China, 8 are water-borne infectious diseases [31]. Wetland parks are close to water sources and are vulnerable to them if they are not properly protected. Therefore, the wetland parks were set as grade I. 

Grade II: the cultural landscape belongs to non-renewable resources. Hangzhou is one of the seven ancient capitals in China. There are abundant cultural relics on the ground and underground. For example, the Liangzhu Cultural Site in the Liangzhu Country Park is more than 4000 years ago. It is one of the most important archaeological sites of the Neolithic Age in the lower reaches of the Yangtze River. Therefore, the heritage parks were set as grade II.

Grade III: mountain type parks mainly include uncertain factors such as traffic, site landform, and geological conditions. Therefore, the mountain parks were set as grade III.

Grade IV: the pastoral park has flat terrain, convenient transportation, high land integration, and relatively complete infrastructure. Therefore, the pastoral parks were set as grade IV.

Grade V: forest park has open terrain and excellent ecological environment. Large dense forests can not only isolate patients, but also provide a good forest rehabilitation environment. Therefore, the forest parks were set as grade V.

### 3.2. Effective Avoidance Area (f*_2_*)

The site selection of emergency medical facilities should consider the lower limit of effective avoidance area. In 2020, with the outbreak of COVID-19, China initiated the level I response to major public health emergencies [32] referring to the construction and use of the Beijing Xiaotangshan, Wuhan Huoshenshan, and Leishenshan hospitals. Based on the data of the new crown pneumonia epidemic, assuming that the emergency medical facility is a single-story building, the effective avoidance area should not be less than 0.047 km^2^ [7]. According to statistics from Hangzhou Country Parks, the effective avoidance area of 29 Country Parks is between 4.31–82.92 km^2^, which all meet the lower limit of land demand. The classification was based on the Natural Breaks (Jenks). Country parks with an effective avoidance area of 4.31–11.72 km^2^ were classified as grade I, 11.73–19.53 km^2^ were classified as grade II, 19.54–32.80 km^2^ were classified as grade III, 32.81–57.93 km^2^ were classified as grade IV, and between 57.94–82.92 km^2^ were classified as grade V.

### 3.3. Spatial Fragmentation Degree (f*_3_*)

Emergency medical facilities require a single, homogeneous, and continuous construction site inside the park. Therefore, a park with a small number of spatial patches and low landscape fragmentation is more suitable as a target site. Landscape fragmentation is usually expressed by the ratio of the number of patches to the corresponding area, that is, Ci = Ni/Ai. The greater the degree of fragmentation, the more dispersed the internal space of the park, and the smaller the area of effective avoidance patches, which is not conducive to the construction of emergency medical facilities and subsequent expansion. The statistical results show that the spatial fragmentation degree of the 29 country parks is between 0.0159–4.868. According to the distribution characteristics of the fragmentation, the Natural Breaks (Jenks) was used for classification, with 0.0159–0.0751 as grade V, 0.0752–0.1677 as grade IV, 0.1678–0.5037 as grade III, 0.5038–1.3322 as grade II, and above 1.3323 as grade I.

### 3.4. Water Source Protection Area (f*_4_*)

According to China’s “Technical guideline for delineating source water protection areas” [27], the land area protection scope of different types of water sources is delimited, of which large reservoirs are the most stringent. For river-type water source areas, the protection scope of the first and second land areas is limited by the horizontal distance between the depth of the coast and the river bank not less than 50 m and 1000 m, respectively. For lake and reservoir water sources, the primary protection includes a land area 200 m above the normal water level at the intake side of the lakes and reservoirs; in the secondary protection, according to the scale of the lakes and reservoirs, it is divided into “nothing outside the primary protection zone”. There are two methods of classification: “Less than 3000 m”, “above the normal water level (outside the primary protection zone), and a horizontal distance of 2000 m”. Combining the characteristics of the water source types in the urban area of Hangzhou, the above divisions are integrated, and the inland area (including water source areas) within 200 m was classified as grade I, 200–500 m as grade II, 500 m–1000 m as grade III, and 1000 m–2000 m as grade IV, >2000 m as grade V.

### 3.5. Wind Direction (f*_5_*)

The Chinese “guidelines for the site selection, design, construction, and operation management of emergency infectious disease hospitals” pointed out that “the location of emergency medical facilities should be located in urban areas that dominate the downwind all the year-round”.

Hangzhou is located to the south of China’s Qinling-Huaihe line. It has a subtropical monsoon climate with a prevailing southwesterly wind. In view of the fact that the dominant wind direction in some cities is not obvious, or there are more than two dominant wind directions, the 16 compass point wind frequency calculation method can be used to quantify the wind frequency in all directions with the core area as the center. Based on the statistics of Hangzhou’s meteorological data, its 16 wind directions and wind frequencies were shown in Table 4. The wind frequency in all directions was mainly distributed below 12%, which was divided into five frequency bands: 10–12.5%, 7.5–10%, 5–7.5%, 2.5–5%, and 0–2.5%, which will correspond to the wind direction of the parks in the upwind area of the city are assigned grades Ⅰ, Ⅱ, Ⅲ, Ⅳ, and Ⅴ in sequence.

### 3.6. Distance from the City Center (f*_6_*)

The Chinese “Design Guidelines for Hospitals for Patients with SARS” and “Guidelines for site selection, design, construction, and operation management of emergency infectious disease hospitals” [26] both pointed out avoidance of “the densely populated areas of the city” as an important principle for site selection. According to the population distribution characteristics of the study area, the boundary of the planning area of the downtown area of Hangzhou is the starting boundary, [33] and the surrounding area is divided into levels I, II, III, IV, and V from near to far with the 10 km as the first level.

### 3.7. Impermeability (f*_3_*)

Medical waste carries a large number of pathogens, heavy metals, and organic pollutants, and can produce a variety of harmful leachate after rainwater and biological hydrolysis [34]. In order to prevent harmful leachate from being washed into the soil with rainwater and thus polluting the groundwater, emergency medical facilities are usually built on sites with high terrain, low groundwater levels and not easy to seep.

Based on the distribution characteristics of soil types in Hangzhou to analyze the impact of groundwater in 29 country parks (Table 5), and the results are as follows: the moisture soil and coastal saline soil area were Grade I, the coarse bone soil (skeleton soil) and yellow soil area were Grade II, the red loam and purple soil area were Grade III, the limestone soil area was Grade IV, and the paddy soil area was Grade V.

### 3.8. Transport Duration (f*_8_*)

In response to the COVID-19 problem, the Chinese government has adopted a centralized isolation and treatment model, which requires high traffic convenience and accessibility to emergency locations. During this period, a total of 7 designated treatment hospitals were set up in the scope of the study [35]. In the same time period, through the digital map navigation system, using the driving mode, starting from the above 7 designated hospitals (Table 6), the transport duration to each country park was estimated, and the longest transport duration was used for evaluation in accordance with the principle of strict evaluation.

Statistics showed that the longest journey time among the 29 country parks was about 2.5 h. The transport duration was divided into five periods of >140 min, 120–140 min, 100–120 min, 80–100 min, and <80 min, which are divided into grade I, II, III, IV, and V in sequence.

To sum up, the classification level of each influencing factor is shown in Table 7.

## 4. Impact Factor Evaluation and Analysis

According to the research on the evaluation method of the above impact factors, the evaluation results of the 29 country parks selected in this paper were as follows (Figure 5):

### 4.1. Park Type Evaluation

Based on statistics on the types of country parks (Table 8), the results showed that among the 29 country parks, there were 5 in grade I, 3 in grade II, 13 in grade III, 5 in grade IV, and 3 in grade V. As Hangzhou has a forest coverage rate of 66.85% and rich cultural relics, some types of country parks overlap, such as Tuankou country park, Dongqiao Country Park and other 10 country parks that have both forest and mountain types; Zhongtai-Xianlin country park is rich in forest resources and overlaps with the key protected areas of ancient tombs in Gudang-Laoheshan-Yuhang Street. It was both a forest type and a heritage type. Based on the principle of lower and strict division, country parks with overlapping types were classified as lower-level for evaluation. On the whole, the type of park is a dominant factor in the site selection process.

### 4.2. Effective Avoidance Area Evaluation

According to the statistics of the effective avoidance area (Table 9), 4 of them belong to grade I, 9 were grade II, 6 were grade III, 4 were grade IV, and 6 were grade V. It is worth noting that even the smallest Qiaosibei Country Park has an effective avoidance area of 4.31 km^2^, which is close to 100 times the underground limit for emergency medical facilities. Therefore, the effective avoidance area of Hangzhou’s country parks is an advantage factor as a whole.

### 4.3. Spatial Fragmentation Degree Evaluation

Based on the statistics of the spatial fragmentation degree of country parks (Table 10 & Figure 6), 1 of them was classified as grade I, 6 in grade II, 7 in grade III, 6 in grade IV, and 9 in grade V. On the whole, the park has a high degree of integration, and the spatial fragmentation degree is a dominant factor. The results showed that the overall spatial fragmentation degree and construction scale show a reverse trend, that is, the larger the scale, the lower the fragmentation, so it is particularly harsh for small-scale parks. For example, Qiaosibei Country Park is the smallest park, divided into 21 patches by highways, railways, urban roads, etc., which greatly reduces the effective avoidance area, and the spatial fragmentation degree is as high as 4.868.

### 4.4. Water Source Protection Area Evaluation

According to the statistics of the water source protection area of the park (Table 11 & Figure 7), there were 12 parks in the first-level area, 6 parks in the second-level area, 5 parks in the third-level area, 5 parks in the fourth-level area, and 1 park in the last-level area. It is worth noting that the parks located within 200 m of the water source (first-level area) are close to half of the total. Among them, 8 country parks including Yunhe Xi’an Country Park and Chengbei Yunhe Country Park are built near the waters, which is closely related to the characteristics of the scenery of Jiangnan Water Village in Hangzhou. On the whole, the park is close to the waters, which is an unfavorable factor in site selection.

### 4.5. Wind Direction Evaluation

Based on the statistics of the parks located in the wind direction evaluation area (Table 12 & Figure 8), there were 9 parks in the first-level area, 13 parks in the second-level area, 0 in the third-level area, and 7 in the fourth-level area, 0 in the last area. The dominant wind direction in Hangzhou is more prominent, with northerly winds (N, NNW, NW) prevailing in winter and southwesterly winds (SSW) prevailing in summer. This pattern of concentrated distribution of wind frequencies can highlight the advantages and disadvantages of the site selection target. In addition, when drawing the urban wind direction, the upwind area is a wind belt with a certain width, which is consistent with the width of the corresponding angle in the central city.

Therefore, the sixteen directions of wind belts with the central city area as the core have a certain range of overlap, and the overlapping area adopts the principle of grading from the lowest. For example, in the positive south wind belt of Hangzhou, the three wind frequencies of grade IV, grade III, and grade II overlap in this order, and the overlapping area of grade IV and grade III will be covered by the grade II wind belt. Moreover, there are 22 country parks located in Grade I and Grade II areas, accounting for 73.33% of the total, which is a disadvantage in site selection. On the whole, the wind direction is a disadvantage in the site selection process.

### 4.6. Distance from the City Center Evaluation

Based on the statistics of the distance between country parks and the central urban area (Table 13 & Figure 9), 16 of them were located in the first-level area, 4 in the second-level area, 2 in the third-level area, 4 in the fourth-level area, and 3 in the last area. By comparison, the number of parks in the first-level area exceeds 53% of the total number and is closer to the city. This is in line with the principle of country parks serving the city and the nearest location, but it is not an ideal place for emergency medical facilities, unless the greening isolation protection measures were done in advance by the government.

### 4.7. Impermeability Evaluation

Based on the statistics of the impermeability of country parks (Table 14 & Figure 10), there were 5 country parks located in the first-level area, 1 in the second-level area, 12 in the third-level area, 4 in the fourth-level area, and 7 in the last-level area. On the whole, since emergency medical facilities are laying an anti-leakage layer on the foundation, the depth of groundwater has a relatively small impact on the overall site selection. However, although the number of parks in the Level I and Level II areas is small, they are too shallow and have a high penetration rate. For example, the buried depth of groundwater in Dayuan Country Park is between 0.5–2.5 m, and its permeability coefficient is between 1.0–5.0 m/d, which is not only unfavorable for engineering construction but also unfavorable for epidemic prevention and control.

### 4.8. Transport Duration Evaluation

Based on the statistics of transport duration (Table 15 & Figure 11), there were 3 parks in grade I, 3 in grade II, 11 in grade III, 9 in grade IV, and 3 in grade V. The results showed that 22 parks are more than 90 min away from the designated hospitals for infectious diseases, and the longest driving distance is even more than 150 min. Therefore, the overall length of traffic is an unfavorable factor in site selection.

By comparison, the transport duration is not completely proportional to the traffic distance. Among them, the country parks located in the suburbs of the central city have many traffic nodes and traffic volume, so the advantage of traffic time is not prominent.

## 5. Results 

According to the evaluation results of the eight impact factors, the comprehensive scores of 29 country parks are calculated by Formula (4), and the results are shown in Figure 12:

Among the 29 country parks, Linglong Country Park has the highest comprehensive scores. The park is a mountain type country park with both forest type, high green coverage, good forest enclosure, good ecological environment, and can provide a better healthy environment for patients. The park has an effective avoidance area of 62.99 km^2^, with unified internal land types and high integration, which can provide sufficient expansion space for emergency medical facilities. In addition, the park is far away from the waters and the central urban area and is located at the lower air outlet all the year-round. It is worth noting that the soil type of the area where the park is located is red soil and limestone soil, and the buried depth of groundwater is shallow, ranging from 0.5 to 3.5 m. The anti-seepage treatment under the facility base should be strengthened according to the standard of landfill site. In addition, Silver Lake Country park, Dongqiao country park, Taihuyuan country park, Tuankou Country Park and other comprehensive conditions are also superior, which can be included in the plan as an alternative for the construction of emergency medical facilities.

## 6. Discussion

By analyzing the research results, it can be concluded that:(1)On the premise of a full score of 100 points, the average score is 57.1 points, which is generally at a medium-to-high level. The highest score is 82.9 points, and the lowest score is 26.6 points. The former is 3.12 times the latter, indicating that this assignment method can significantly separate the pros and cons of country parks and has a clear degree of distinction;(2)The high-quality parks in Lin’an District and Fuyang District are relatively more concentrated than others, and there is an indirect correspondence between the pros and cons of parks and their spatial locations. Among the eight impact factors, the two impact factors of distance from the city center and the transport duration are directly related to the spatial location, and the other six impact factors are very minimally affected by the spatial location. Therefore, the spatial location of the park does not play a decisive role in its comprehensive score (Figure 13);(3)Among the top 5 parks, five impact factors such as effective avoidance area, spatial fragmentation degree, distance from the waters, wind direction, and distance from the central urban have outstanding advantages. The effective avoidance area of the 5 parks is greater than 50 km^2^, and the site integration is relatively high; the water source protection area is far, and the spacing is greater than 1 km; 4 of the parks are located in a wind direction of level IV, and 4 parks are more than 40 km away from the central city;(4)The validity advantage of the impact factor of transport duration is not obvious, and there are Ⅰ, Ⅱ, and Ⅲ grades in the top 5 country parks.

## 7. Conclusions

In view of the characteristics of low and sudden public health emergencies, the disaster prevention function of the country park can be used to carry out the corresponding emergency plan planning through the method of combining disaster relief, so as to alleviate the difficulties of scientific site selection and timeliness of construction of emergency medical facilities. Through comparison, it is found that country parks are highly consistent with the demand for emergency medical facilities in terms of site selection, scale, environment, infrastructure, coverage, etc. The inclusion of country parks in the emergency plan for public health emergencies can greatly improve the speed of emergency medical facilities construction and reduce the harm of infectious diseases to the population and the environment to a certain extent by means of early rational location and layout, pre-exploration and laying of pipe networks, strengthening sewage and garbage treatment systems, and rational planning of health functions.

The country park selected as the target of emergency medical facilities should be upgraded and transformed according to the construction standard of infectious disease emergency medical facilities: (1) a flat space with the corresponding scale is reserved as the construction site of emergency medical facilities in the event of the COVID-19 outbreak. This space is preferred to sunny slopes, high terrain, low groundwater level, open and flat lawns, or wasteland; (2) carry out corresponding surveys on the selected sites to make the geological conditions clear, lay necessary infrastructure pipelines such as water, electricity, gas, and communication in the reserved sites, improve the standards of the sewage treatment system in the park, and add harmless waste treatment facilities to meet the requirements of the “Technical Specification for Centralized Disposal of Medical Wastes”, so as to save construction time in case of the outbreak of COVID-19 epidemic; (3) pay attention to the selection and configuration of healthy plants, create a microclimate on the site, and provide a good rehabilitation environment for patients; (4) the drawings of emergency medical facilities shall be prepared in advance to preset the construction process, and the relevant facilities and material supply departments shall be included in the plan.

This paper analyzes 8 influencing factors to determine the location of emergency medical facilities, lacking consideration of the construction cost of emergency medical facilities, the purchase cost of emergency resources, and the transportation cost; In addition, this study screened the influencing factors based on the unique geographical environment, climate characteristics and construction status of Hangzhou. Due to the large differences in conditions in different regions, each factor has different degrees of influence on the location of emergency medical facilities. The types of influencing factors can be increased or decreased in combination with the local situation, so as to give play to the favorable conditions of the destination and avoid adverse factors.

## Figures and Tables

**Figure 1 ijerph-19-15876-f001:**
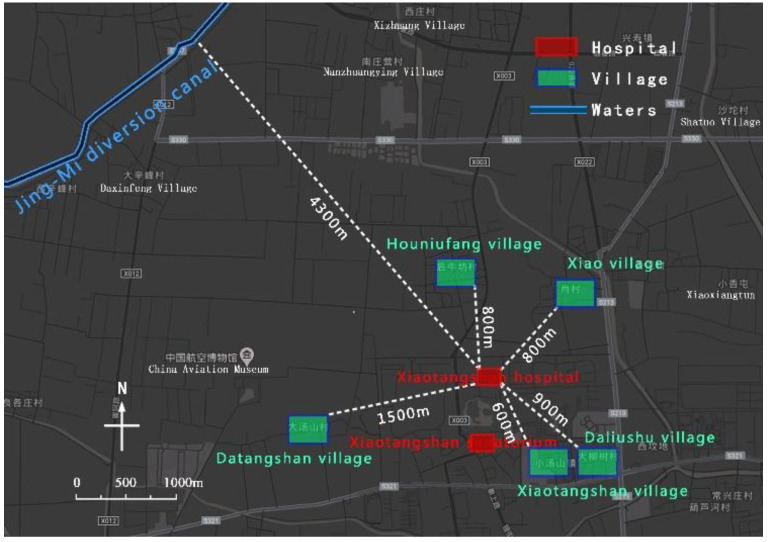
Site environment of Xiaotangshan hospital in 2003.

**Figure 2 ijerph-19-15876-f002:**
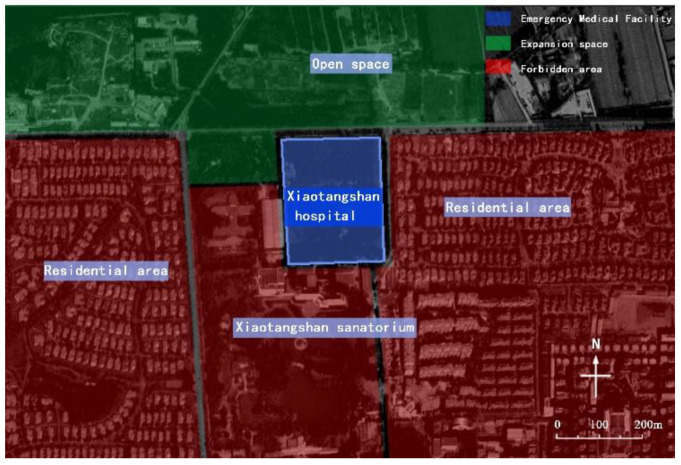
Site environment of Xiaotangshan hospital in 2020.

**Figure 3 ijerph-19-15876-f003:**
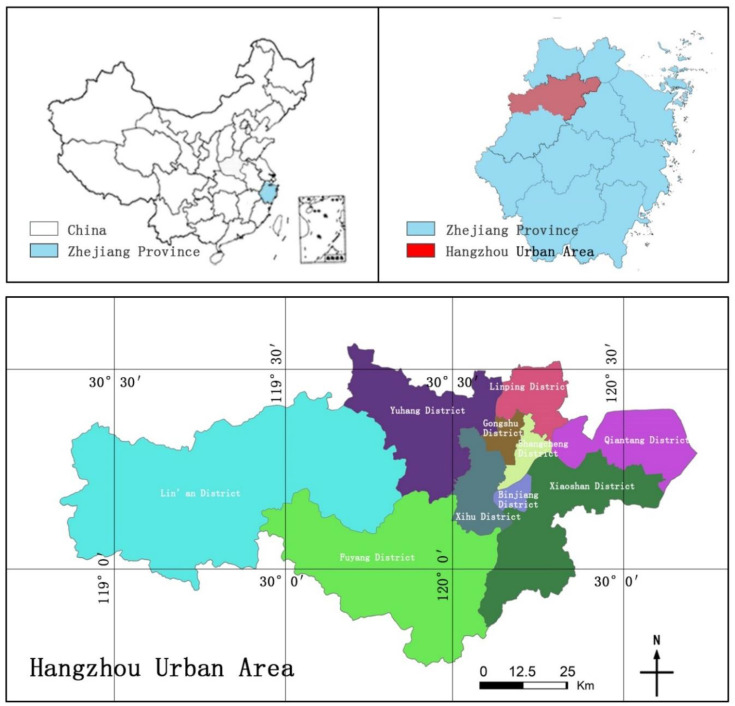
The location of Hangzhou Urban.

**Figure 4 ijerph-19-15876-f004:**
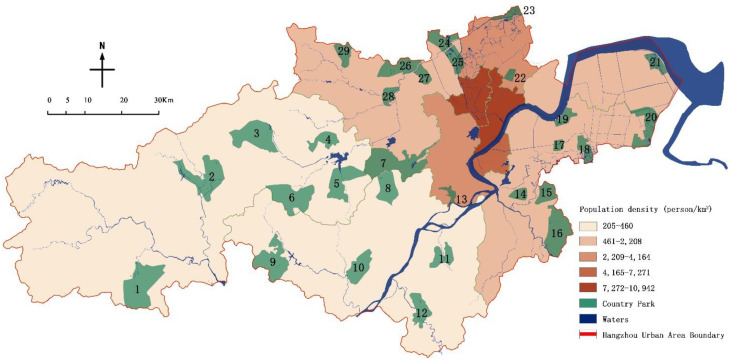
Population density map of Hangzhou.

**Figure 5 ijerph-19-15876-f005:**
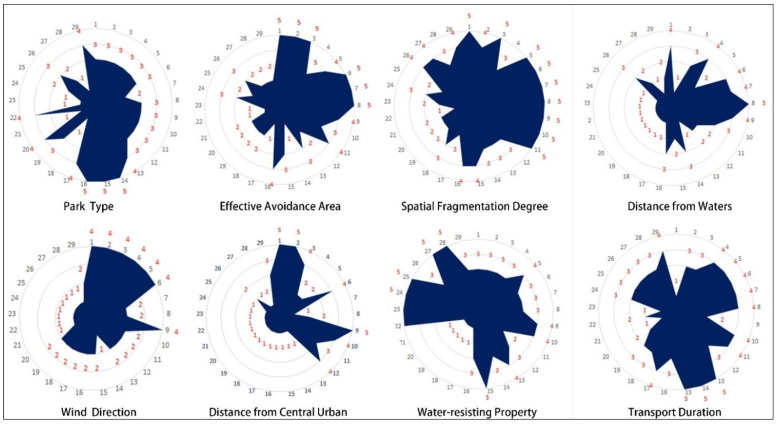
Scores of the impact factors.

**Figure 6 ijerph-19-15876-f006:**
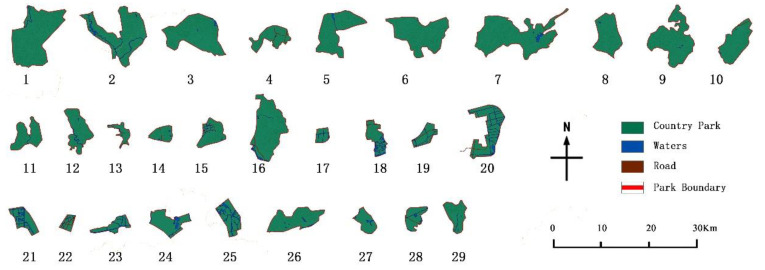
The spatial fragmentation of country parks.

**Figure 7 ijerph-19-15876-f007:**
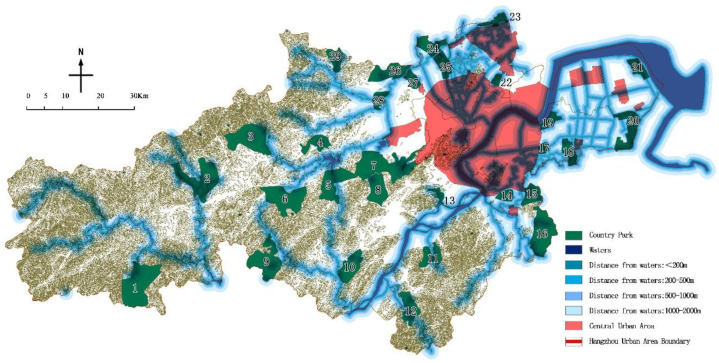
Distances of country parks from waters.

**Figure 8 ijerph-19-15876-f008:**
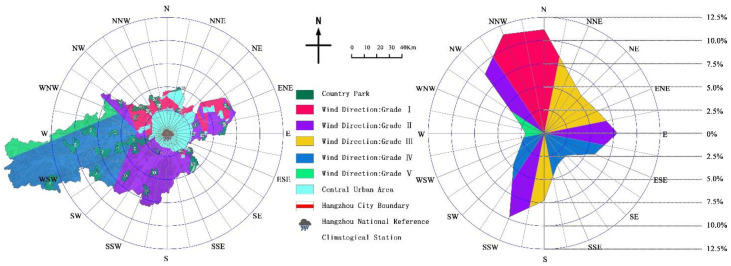
The wind direction distribution of country parks.

**Figure 9 ijerph-19-15876-f009:**
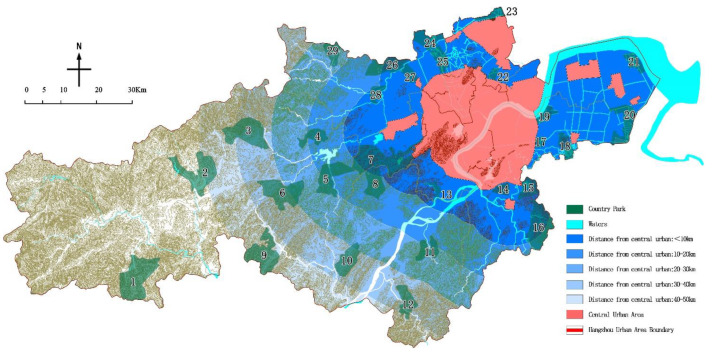
The distance from city center of the country parks.

**Figure 10 ijerph-19-15876-f010:**
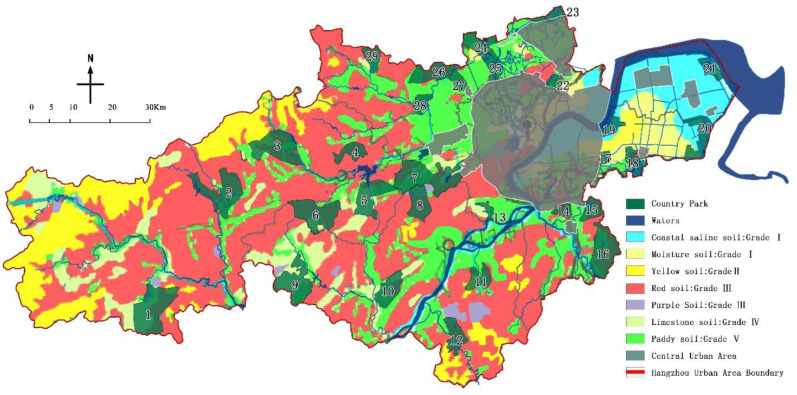
The impermeability distribution of the country parks.

**Figure 11 ijerph-19-15876-f011:**
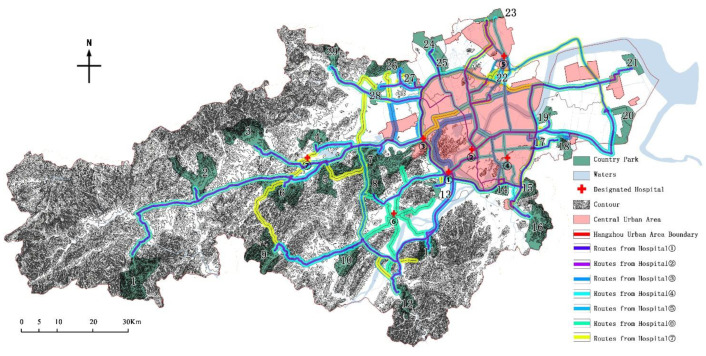
The traffic routes of the country parks to designated hospital.

**Figure 12 ijerph-19-15876-f012:**
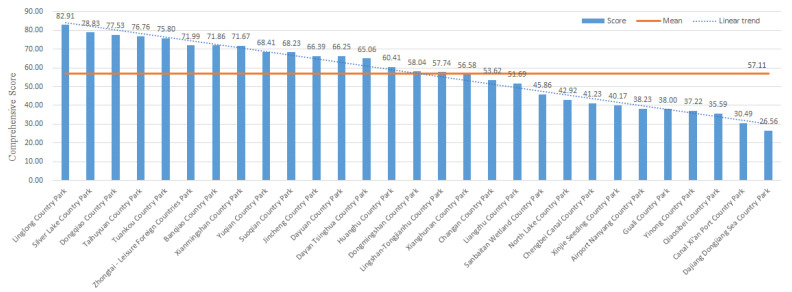
Comprehensive score of the country parks.

**Figure 13 ijerph-19-15876-f013:**
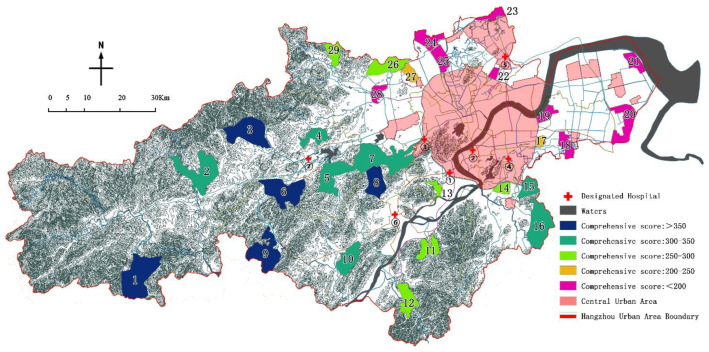
Visualization of the ranking and classification of the country parks.

**Table 1 ijerph-19-15876-t001:** Statistics of country parks in Hangzhou Urban.

Country Park	Number	Park Type	Effective Avoidance Area (hm^2^)	Spatial Fragmentation Degree	Transport Duration (min)
Tuankou Country Park	1	Mountain & Forest type	80.61	0.0372	156
Yuqian Country Park	2	Mountain & Forest type	68.92	0.1451	108
Taihuyuan Country Park	3	Mountain & Forest type	67.61	0.0444	114
Jincheng Country Park	4	Mountain & Forest type	23.64	0.2538	84
Banqiao Country Park	5	Mountain & Forest type	46.52	0.0645	84
Linglong Country Park	6	Mountain & Forest type	62.99	0.0159	90
Zhongtai-Leisure Foreign Countries Park	7	Forest & Heritage type	82.92	0.0362	84
Silver Lake Country Park	8	Mountain & Forest type	62.92	0.0318	84
Dongqiao Country Park	9	Mountain & Forest type	57.93	0.0173	126
Xianmingshan Country Park	10	Pastoral & Mountain type	32.80	0.061	96
Dayuan Country Park	11	Mountain & Forest type	39.92	0.0751	90
Changan Country Park	12	Mountain & Forest type	15.35	0.2605	114
Lingshan-Tongjianhu Country Park	13	Pastoral type	27.76	0.3602	72
Xianghunan Country Park	14	Forest type	11.72	0.4267	78
Suoqian Country Park	15	Forest type	27.44	0.1093	78
Dayan Tsinghua Country Park	16	Forest type	54.82	0.1095	102
Xinjie Seeding Country Park	17	Pastoral type	7.61	0.9193	96
Guali Country Park	18	Wetland type	15.88	0.5037	114
Airport Nanyang Country Park	19	Mountain type	14.16	0.9888	102
Yinong Country Park	20	Pastoral type	19.52	1.3322	132
Dajiang Dongjiang Sea Country Park	21	Wetland type	14.87	0.874	156
Qiaosibei Country Park	22	Pastoral type	4.31	4.868	138
Canal Xi’an Port Country Park	23	Wetland & Heritage type	10.57	1.1348	144
Sanbaitan Wetland Country Park	24	Wetland type	23.28	0.2577	102
Chengbei Canal Country Park	25	Humanistic type	17.62	0.8513	102
Dongmingshan Country Park	26	Mountain type	29.69	0.1347	108
Liangzhu Country Park	27	Humanistic type	19.53	0.1536	108
North Lake Country Park	28	Wetland type	14.56	0.412	108
Huanghu Country Park	29	Pastoral type	17.89	0.1677	96

**Table 2 ijerph-19-15876-t002:** Table of principal component load factors.

Impact Factor	Principal Component		
	*F* _1_	*F* _2_	*F* _3_
Park type (*f*_1_)	0.246	0.760	−0.311
Effective avoidance area (*f*_2_)	0.888	−0.136	0.089
Spatial fragmentation degree (*f*_3_)	0.898	0.033	0.238
Water source protection area (*f*_4_)	0.790	0.113	0.125
Wind direction (*f*_5_)	0.814	−0.152	−0.275
Distance from the city center (*f*_6_)	0.771	−0.333	−0.114
Impermeability (*f*_7_)	−0.037	0.037	0.915
Transport duration (*f*_8_)	0.231	0.854	0.132

**Table 3 ijerph-19-15876-t003:** Eigenvalues and cumulative contribution rates of principal components.

Principal Component	Initial Eigenvalues		
	Eigenvalue (*λ*)	Variance Contribution (%)	Cumulative Contribution (%)
*F* _1_	3.590	44.877	44.877
*F* _2_	1.475	18.434	63.311
*F* _3_	1.120	13.999	77.311
*F* _4_	0.687	8.581	85.892
*F* _5_	0.496	6.196	92.088
*F* _6_	0.300	3.747	95.836
*F* _7_	0.241	3.017	98.853
*F* _8_	0.092	1.147	100.000

**Table 4 ijerph-19-15876-t004:** Annual frequency of wind directions in Hangzhou (%).

Wind Direction	N	NNE	NE	ENE	E	ESE	SE	SSE
Frequency	11.19	7.02	5.99	6.45	8.13	4.96	3.24	3.63
**Wind Direction**	**S**	**SSW**	**SW**	**WSW**	**W**	**WNW**	**NW**	**NNW**
Frequency	7.22	9.63	4.78	2.77	2.11	2.40	8.96	11.52

**Table 5 ijerph-19-15876-t005:** Statistical of impermeability degree.

Number	Soil Type	Groundwater Depth (m)	Permeability Coefficient (m/d)	Impermeability
1	Red Soil	0.5–2.5	0.5–1.5	0.3333–5
2	paddy soil	0.5–3.5	0.05–0.5	1–70
3	moisture soil	−0.5–2	0.1–1.5	−0.3333–20
4	Coastal saline soil	−0.5–2	0.05–1.0	−0.5–40
5	purple soil	1–4.5	0.5–3.0	0.3333–9
6	limestone soil	0.5–3.5	0.1–1.0	0.5–35
7	skeleton soil	0.5–4	50–150	0.0033–0.08
8	yellow soil	0.5–2.5	1.0–5.0	0.1–2.5

**Table 6 ijerph-19-15876-t006:** Designated hospitals for COVID-19 treatment in Hangzhou Urban.

The Designated Hospital of Hangzhou Urban	Number
The First Affiliated Hospital, Zhejiang University School of Medicine	①
The Children’s Hospital, Zhejiang University School of Medicine	②
Xixi Hospital of Hangzhou	③
The First People’s Hospital of Xiaoshan Hangzhou	④
The First People’s Hospital of Yuhang Hangzhou	⑤
The First People’s Hospital of Fuyang Hangzhou	⑥
The First People’s Hospital of Lin’an Hangzhou	⑦

**Table 7 ijerph-19-15876-t007:** Grading criteria for influencing factors.

Variables	Park Type	Effective Avoidance Area (km^2^)	Spatial Fragmentation Degree	Water Source Protection Area (m)	Wind Direction	Distance from the City Center (km)	Impermeability	Transport Duration (min)
I	wetland park	4.31–11.72	>1.3323	<200	10–12.5%	<10	moisture soil, coastal saline soil	>140
II	heritage park	11.73–19.53	0.5038–1.3322	200–500	7.5–10%	10–20	coarse bone soil (skele-ton soil), yellow soil	120–140
III	mountain park	19.54–32.80	0.1678–0.5037	500–1000	5–7.5%	20–30	red loam, purple soil	100–120
IV	pastoral park	32.81–57.93	0.0752–0.1677	1000–2000	2.5–5%	30–40	limestone soil	80–100
V	forest park	57.94–82.92	0.0159–0.0751	>2000	0–2.5%	40–50	paddy soil	<80

**Table 8 ijerph-19-15876-t008:** The results of park type assignment of country parks.

Park Number	1	2	3	4	5	6	7	8	9	10	11	12	13	14	15
Park type assignment	III	III	III	III	III	III	II	III	III	III	III	III	IV	V	V
**Park Number**	**16**	**17**	**18**	**19**	**20**	**21**	**22**	**23**	**24**	**25**	**26**	**27**	**28**	**29**	
Park type assignment	V	IV	Ⅰ	III	IV	Ⅰ	IV	Ⅰ	Ⅰ	II	III	II	Ⅰ	IV	

**Table 9 ijerph-19-15876-t009:** The results of effective avoidance area assignment of country parks.

Park Number	1	2	3	4	5	6	7	8	9	10	11	12	13	14	15
Effective avoidance area	V	V	V	III	IV	V	V	V	IV	III	IV	II	III	Ⅰ	III
**Park Number**	**16**	**17**	**18**	**19**	**20**	**21**	**22**	**23**	**24**	**25**	**26**	**27**	**28**	**29**	
Effective avoidance area	IV	Ⅰ	II	II	II	II	Ⅰ	Ⅰ	III	II	III	II	II	II	

**Table 10 ijerph-19-15876-t010:** The results of spatial fragmentation degree assignment of country parks.

Park Number	1	2	3	4	5	6	7	8	9	10	11	12	13	14	15
Spatial fragmentation degree assignment	V	IV	V	III	V	V	V	V	V	V	V	III	III	III	IV
**Park Number**	**16**	**17**	**18**	**19**	**20**	**21**	**22**	**23**	**24**	**25**	**26**	**27**	**28**	**29**	
Spatial fragmentation degree assignment	IV	II	III	II	II	II	Ⅰ	II	III	II	IV	IV	III	IV	

**Table 11 ijerph-19-15876-t011:** The results of water source protection area of country parks.

Park Number	1	2	3	4	5	6	7	8	9	10	11	12	13	14	15
Water source protection area assignment	IV	Ⅰ	III	IV	II	IV	IV	V	IV	III	II	II	Ⅰ	III	II
**Park Number**	**16**	**17**	**18**	**19**	**20**	**21**	**22**	**23**	**24**	**25**	**26**	**27**	**28**	**29**	
Water source protection area assignment	III	Ⅰ	Ⅰ	Ⅰ	Ⅰ	Ⅰ	Ⅰ	Ⅰ	Ⅰ	Ⅰ	III	II	Ⅰ	II	

**Table 12 ijerph-19-15876-t012:** The results of wind direction assignment of country parks.

Park Number	1	2	3	4	5	6	7	8	9	10	11	12	13	14	15
Wind direction assignment	IV	IV	IV	IV	IV	IV	II	II	IV	II	II	II	II	Ⅰ	II
**Park Number**	**16**	**17**	**18**	**19**	**20**	**21**	**22**	**23**	**24**	**25**	**26**	**27**	**28**	**29**	
Wind direction assignment	II	II	II	II	II	I	I	I	I	I	I	I	I	II	

**Table 13 ijerph-19-15876-t013:** The results of distance assignment of country parks from city center.

Park Number	1	2	3	4	5	6	7	8	9	10	11	12	13	14	15
Distance from city center assignment	V	V	IV	II	II	IV	I	II	V	IV	III	IV	I	I	I
**Park Number**	**16**	**17**	**18**	**19**	**20**	**21**	**22**	**23**	**24**	**25**	**26**	**27**	**28**	**29**	
Distance from city center assignment	I	I	I	I	I	I	I	I	I	I	II	I	I	III	

**Table 14 ijerph-19-15876-t014:** The assignment results of the impermeability of country parks.

Park Number	1	2	3	4	5	6	7	8	9	10	11	12	13	14	15
Impermeability assignment	III	III	III	III	IV	III	III	III	IV	IV	II	III	IV	III	V
**Park Number**	**16**	**17**	**18**	**19**	**20**	**21**	**22**	**23**	**24**	**25**	**26**	**27**	**28**	**29**	
Impermeability assignment	III	I	I	I	I	I	V	V	V	V	III	V	V	III	

**Table 15 ijerph-19-15876-t015:** The results of transport duration assignment of country parks.

Park Number	1	2	3	4	5	6	7	8	9	10	11	12	13	14	15
Transport duration assignment	I	III	III	IV	IV	IV	IV	IV	II	IV	IV	III	V	V	V
**Park Number**	**16**	**17**	**18**	**19**	**20**	**21**	**22**	**23**	**24**	**25**	**26**	**27**	**28**	**29**	
Transport duration assignment	III	IV	III	III	II	I	II	I	III	III	III	III	III	IV	

## Data Availability

Data is contained within the article.

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
