# Peer review of "Country Parks as Sites of Emergency Medical Facility: A Case of COVID-19 in Hangzhou, China"

_ijerph, 2022, doi:10.3390/ijerph192315876_

Round 1
Reviewer 1 Report
COVID 19 has highlighted caveats in existing health planning systems across counties, and provided directions towards scope for improvements in the health system. Using country parks for emergency response is extremely useful for any large city globally that experienced a huge surge of COVID 19 cases. Thus, the authors investigating use of parks for emergency is very timely and important topic. This study also adopts a sophisticated method. That said, following are some of the points that, according to me, require attention:
1. The authors justify the need for this research by explaining the caveats in the site selection process for Xiaotanhshan Hospital was used. It is not clear if the hospital worked as emergency medical facilities or a permanent site. If the latter is true, how can the use of parks as an emergency cite can be justified by highlighting the limitations in the site selection for a permanent hospital. The scope of this research needs further justification.
2. In Section 1.2 of Page 3, the authors introduced the research site. Some statistics on the impact of COVID-19 in the region will better justify the selection of the study area.
3. Table 1 lists a number of variables explaining the characteristics of the parks. There needs to be some definition of what each of the variable means. The definitions can be provided in the beginning of Section 2.2, where the authors mention the measures in line 141 and 142.
4. It will be helpful to have an idea of population density in the study area relative to the park locations. The authors can consider mapping the population density instead of contour in Figure 4, and display the country parks.
5. In Section 2.2 paragraph two, the authors explain the capacities of PCA. However, the purpose of using PCA in this research should be clearly stated in the very beginning of the paragraph. Also, the description in this section requires substantial clarity. Similarly, the description of impact factor calculation requires simplification. The method is not very clear at the present state.
6. In Section 3, a table summarizing grading criteria of the parks comprising all impact factors will be helpful for the readers.
7. The “Results and Conclusion” section requires substantial improvement. To better communicate the results, the “Results and Conclusion” section could be divided into two sections “Results” and “Discussion and Conclusion”. The narrative should be substantially improved in order to deliver the key messages. The limitations of the research should also be highlighted.
8. This article needs substantial language editing. There are need for editorial corrections. For example, In line 129, instead of “arcgis 10.5”, it should be ArcGIS 10.5. Also, the sentence structure needs to be revisited. The sentences are long, complex, and hard to understand. Further simplification is need for the readers to understand.
Reviewer 2 Report
The paper is about searching for a site for emergency medical facilities, such as during the COVID-19 pandemic. You should use COVID-19, rather than the obsolete novel coronavirus pneumonia. The introduction can be substantially shortened as they refer mainly to China and is of little interest to international readers. What is curiously missing is the literature review, e.g., similar hunt for country parks as emergency medical facilities in other countries and the different variables. Be consistent with terminology, e.g., use the standard Xi as variables, and don't call them impact factors. I don't know why the variables are labelled mi in Table 2 but fi in Section 3. It is also helpful to rename some of the variables, e.g., "effective avoidance area" is confusing compared to "Useable area". Similarly, "Distance from central urban" should be "Accessibility" or "Distance from urban (or suburban) center". Use Table 1 and not Tab 1. The weight model is unclear and appears out of nowhere. If the purpose of PCA is to reduce dimensionality, then the loadings require interpretation.
Round 2
Reviewer 2 Report
This is a better version.